# Correlation of pitching velocity with anthropometric measurements for adult male baseball pitchers in tryout settings

Jyh How Huang[1], Szu-Hua Chen[2], Chih Hui Chiu[3]*

1 Sport Information and Communication Department, National Taiwan University of Sport, Taichung, Taiwan, 2 Department of Physical Therapy, Ithaca College, Ithaca, NY, United States of America, 3 Graduate Program in Department of Exercise Health Science, National Taiwan University of Sport, Taichung, Taiwan

* loveshalom@hotmail.com

**Data Availability Statement:** The dataset is uploaded and shared publicly on Zenodo https://zenodo.org/record/5796004#.YcHpkdBByUk DOI: 10.5281/zenodo.5796004 https://doi.org/10.5281/zenodo.5796004.

## Abstract

Several studies have investigated factors influencing baseball pitching velocity. However, some measurements require expensive equipment, and some tests need familiarity to perform well. In this study, we adopted field tests executed using affordable equipment in a tryout event for a professional baseball team in Taiwan, 2019. We use half day to test 64 players, and the result of measurement are used to develop a model for predicting pitching velocity of amateur adult pitchers (age: 23.9 ± 2.8 years; height: 180.3 ± 5.9 cm; weight: 81.4 ± 10.9 kg). The measurements and tests in tryout settings should be easy to implement, take short time, do not need high skill levels, and correlate to the pitching velocity. The outcome measures included maximum external shoulder rotation, maximum internal shoulder rotation, countermovement jump (CMJ) height, 20-kg loaded CMJ height, 30-m sprint time, height, age, and weight tests. Multiple regression indicated a moderate correlation between these tests and pitching velocity (adjusted $R^2$ = 0.230, $p$ = 0.0003). Among the measures, the ratio of loaded CMJ to CMJ, ratio of first 10-m sprint time to 30-m sprint time, and height were significant contributors to pitching velocity. Overall, these measures explained 23% of the variance in the predicted pitching velocity. These field tests can be adopted in tryout events to predict a prospect's potential and to identify underestimated players. Coaches can obtain an expectation of a pitcher's performance by comparing his pitching velocity with the predicted value derived from the statistical model presented herein, and the room of growth by comparing his current strength to average strength growth after being drafted and trained with professional coaches.

## Introduction

Pitching velocity is a key indicator of pitching performance. To determine strategies for improving pitching velocity, numerous studies have investigated the effects of elbow and shoulder kinetic parameters [1,2]. However, whether improvements in elbow and shoulder kinetic parameters can increase pitching velocity is unclear. Oi et al. suggested that the

**Funding:** This work is sponsored by Ministry of Science and Technology, Taiwan Award Number: 109-2622-H-028-001 | Recipient: JyhHow Huang The funders had no role in study design, data collection and analysis, decision to publish, or preparation of the manuscript.

**Competing interests:** The authors have declared that no competing interests exist.

pitching velocity of US pitchers was faster than that of Japanese pitchers; however, such a finding might be affected by anthropometric measures because the Japanese pitchers demonstrated larger shoulder horizontal adduction torque when the parameters were normalized to body weight and height [3]. Therefore, elbow and shoulder kinetic parameters during pitching movement are not the only factors influencing pitching velocity.

Pitching velocity has been widely explored. It could depend on several factors, including talent, mechanics, and strength and conditioning. Studies have reported that better temporal parameters and kinematic parameters often infer to faster pitching velocity [4,5]. Research also revealed that resistance training can increase pitching velocity [6]. Additionally, a study investigating the correlation between lower-extremity ground reaction forces and pitching mechanics determined that such forces are correlated with pitching velocity [7,8]. Lehman et al. performed lower-body field tests and correlated the results with pitching velocity [9]. Studies exploring predictors of the pitching velocity of pitchers have reported different findings. For example, a study investigated predictors of pitching velocity by recruiting participants aged 14.7 ± 2.6 years; the significant predictive factors were age, height, weight, body mass index, shoulder external rotation, and total arc of shoulder rotation, and significant kinematic factors were maximum knee height, stride length, knee flexion, foot angle, lead hip flexion, lateral trunk tilt, and hip–shoulder separation [10]. Mercier et al. conducted a thorough review of factors associated with baseball pitching performance and observed that body weight, age, lateral-to-medial jump, medicine ball scoop, standing long jump, 10-m sprint, and grip strength were significantly associated with pitching velocity [11]. For physical tests, the pitching kinetic energies were significantly correlated with height ($R = 0.870$), standing long jump ($R = 0.850$), sprint time ($R = -0.759$) and grip strength ($R = 0.906$ for right, and 0.894 for the left hand) in 164 youth baseball players aged 6.4–15.7 years[12]. Hoffman et al. compared anthropometric and performance variables in professional baseball players and the regression analysis showed that performance measures accounted for 25–31% of the variance in baseball batting performance[13]. Watanabe et. al. measured female baseball players' physical fitness and found the that lower-limb power contributes to baseball hitting outcomes at a low to moderate level ($R^2 = 0.144$–0.315) [14], which is close to our finding for pitchers($R^2 = 0.230$). Priest et al. conducted modified 60-yard run shuttle in their NCAA baseball tryouts, which is closest to our research, but did not correlate the measurements with pitching performance [15].

Only few studies have focused on the use of affordable, inexpensive tests to quantify potential pitching velocity in tryout events. I.e., how accurately can we predict a pitcher's pitching velocity with affordable field test equipment in short period of time, with easy to execute tests? From past researches, we picked several tests which are quick and easy to implement. Past study has found the ratio of CMJ and loaded CMJ correlates to an athlete's ability to produce higher peak and instantaneous forces [16], and is verified as one of the variables that correlates to pitching velocity in our research. Sprinting time of different distances, mostly 10 meters, 30 meters, and 60 yards, to the performance of baseball players have be discussed in several researches [12,17,18]. We chose 30-meter sprint for the tryout event, while first 10-meter's time is also measured, and ratio of 10 to 30 meters calculated. Deriving an equation for predicting pitching velocity by using anthropometric measurements and inexpensive field tests in tryout settings can enable professional teams, coaches, and scouts a quick and easy way to identify underestimated pitchers who might not demonstrate fast pitching velocity due to poor mechanics or pichers with very good mechanics but lack of adequate strength training and conditioning.

Accordingly, the objective of this study was to examine whether pitching velocity would be correlated with the results of anthropometric measurements and field tests that can be conducted quickly with affordable equipment. By analyzing data collected during a tryout event

for a professional baseball team in Taiwan in 2019, we obtained predictive models that can be used to identify pitchers with potentially high pitching velocities.

## Methods

On the basis of procedures described by previous studies, we selected maximum external shoulder rotation, maximum internal shoulder rotation [5,19], age, height, weight [20], 30-m sprint time, and first 10-m sprint time [9,21] as the outcome measures based on significance, ease of execution, and cost of testing equipment. The ability to produce a relatively high peak force (PF), instantaneous force (IF), and isometric rate of force development (IRFD) was related to smaller differences between 20-kg loaded and unloaded jump heights [9]. Both loaded and unloaded countermovement jumps (CMJs) were significantly correlated with PF, peak power, and net impulse [22]. Accordingly, we included loaded and unloaded jump tests to assess player PF, IF, and IRFD and to predict pitching velocity.

### Participants

A total of 64 Asian adult male pitchers (age: 23.9 ± 2.8 years; height: 180.3 ± 5.9 cm, weight: 81.4 ± 10.9 kg) were recruited during a tryout event in 2019 for a professional baseball team. During this event, the pitchers underwent several anthropometric measurements and field tests. This study was approved by the Antai Tian-Sheng Memorial Hospital Ethics Committee (IRB reference number: 19-024-B). All participants were informed of the study purpose and data handling and provided written consent before participating in the study. Nine of the participants were excluded because they demonstrated a pitching velocity of less than 130 km/h.

This study was designed to determine whether the selected anthropometric measurements and performance in field tests were correlated with pitching velocity. The participants were familiarized with the testing protocols and underwent a standardized 10-min dynamic warm-up process, including walking knee lift, walking over and under, leg cradle, lunge walk, lunge with a twist, high-knee walk, skip, and run. The anthropometric measurements of interest were body height, body weight (both measured with Tanita WB-380H; height output scale: 1 cm; weight graduation: 0.1 kg), and the passive range of motion (ROM) of internal and external shoulder rotation. The field tests comprised CMJ, loaded CMJ, and 30-m sprint tests.

### ROM measurement

A joint protractor was used to measure the passive internal and external rotation of the shoulder joints on the pitching side of the participants when lying [23]. Each movement was measured twice, and the movement was tested thrice if the difference between the two results was greater than 5˚. Of the three test results, the two results within a 5˚ range were averaged for further analysis. The rotation on the nonpitching side was not measured.

### Loaded and unloaded CMJ

The loaded and unloaded CMJ tests were conducted using Gymaware (GymAware Lite v2.10; distance resolution: 0.3 mm; variable rate sampling base rate: 115,200 Hz) [24]. For the unloaded CMJ, each of the participants was instructed to keep their hands on the waist, squat until their thighs were parallel to the floor, and then explode off the ground to maximum height without pausing. A harness was attached to the waist of the participant during the test. The loaded CMJ involved a similar protocol to the unloaded CMJ, but the participant was asked to carry a 10-kg dumbbell in each hand (total weight 20 kg) and have their hands hang on the sides of their body. The loaded and unloaded CMJ protocols were executed twice, with

1 min of rest between them. A third test was conducted only if the difference between the results of the two tests was greater than 5 cm. The measured heights were averaged as the final result.

## Speed test

A timing gate system (Witty Wireless Training Timer; resolution: 1/25,000 s) [25] was used to measure performance in the 30-m sprint test, in which 0–10- and 0–30-m sprint times were recorded. During the test, the participants were asked to keep their feet behind the starting line and then sprint for 30 m at full intensity. All participants use standing start, and each participant must complete two sprint tests, of which the faster sprint time was used for further analysis.

## Velocity

During the tryout, pitching velocity was measured using Rapsodo Pitching 2.0. Rapsodo Pitching has been adapted by all Major League Baseball teams for measuring pitching velocity, movement, spin axis, and spin rate [26]. The Rapsodo unit is setup according to its manual, and was not moved or changed during the tryout. Participants warm up in a manner analogous to a game depending to their personal habits, including stretching and other drills. They know their try out ID number, and the numbers are called in order, so they can throw warm-up pitches based on their personal habits to try to perform best during the test. During the test, they are asked to throw 5 to 10 fastballs, and then some more breaking balls upon the coaches' and scouts' requests. The measurements and field test results are listed in Table 1.

## Statistical analysis

Descriptive values are expressed as mean (M) ± SD. A linear stepwise multiple regression analysis was performed using the R with step function in {stats} package to investigate the correlations of the five anthropometric measurements, three physical parameters, and various field test performance results with pitching velocity and to determine an optimal set of parameters for predicting pitching velocity. These parameters included body height, body weight, age, maximum international rotation, external rotation of throwing shoulder, 30-m sprint time ($sprint_{30}$), first 10-m sprint time ($sprint_{10}$), unloaded CMJ ($CMJ_{unloaded}$), and loaded CMJ ($CMJ_{loaded}$). The ratio of $sprint_{10}$ to $sprint_{30}$ ($sprint_{ratio}$) and the ratio of $CMJ_{loaded}$ to $CMJ_{unloaded}$ ($CMJ_{ratio}$) were calculated. The variance inflation factor (VIF) was used to test for

**Table 1. Anthropometric, physical parameters and field test results.**

|  | Mean | SD |
|---|---|---|
| Age (yrs) | 23.9 | 2.8 |
| Height (cm) | 180.3 | 5.9 |
| Weight (kg) | 81.4 | 10.9 |
| $ER_{ROM}$ (°) | 107.7 | 7.8 |
| $IR_{ROM}$ (°) | 52.5 | 11.3 |
| $Sprint_{10}$ (sec) | 1.82 | 0.08 |
| $Sprint_{30}$ (sec) | 4.05 | 0.17 |
| $Sprint_{ratio}$ | 0.45 | 0.02 |
| $CMJ_{loaded}$ (cm) | 35.49 | 5.29 |
| $CMJ_{loaded}$ (cm) | 42.97 | 2.83 |
| $CMJ_{ratio}$ | 0.83 | 0.07 |

multicollinearity. Results are expressed with adjusted $R^2$, residual standard error (RSS), and regression equations. The alpha level was set to .05.

## Results

In the stepwise multiple linear regression analysis, $sprint_{10}$, weight, $ER_{ROM}$, $IR_{ROM}$, $sprint_{30}$, and age were eliminated in this order. Of the remaining variables, $CMJ_{loaded}$, $CMJ_{unloaded}$, $CMJ_{ratio}$, $sprint_{ratio}$, and height were selected on the basis of the sum of the squares.

The variance inflation factor was used to test for multicollinearity, and $CMJ_{unloaded}$ and $CMJ_{loaded}$ were eliminated in the process. Therefore, the regression indicated an overall significant correlation of height, $CMJ_{ratio}$, and $Sprint_{ratio}$ with pitching velocity ($F_{(4, 59)} = 7.26$, $p < .001$, adjusted $R^2 = .23$), indicating that these parameters could explain approximately 23% of the variance in pitching velocity. Furthermore, examining individual predictors revealed that height ($t = 2.82$, $p = .007$), $sprint_{ratio}$ ($t = 2.98$, $p = .004$), and $CMJ_{ratio}$ ($t = 2.35$, $p = .022$) were significant predictors in the model. The following model was thus derived for predicting pitching velocity:

Velocity = 53.22 + 0.21 (height) + 14.3 (cmjWtoNW) + 75.03 (sprint10to30) + $\epsilon$, **adjusted $R^2$ = 0.230, RSS = 3.57, $p$ = 0.0003113**

The coefficients with $p$ values for each parameter of the equation are listed in Table 2.

## Discussion

This study investigated the correlation of anthropometric measurements and field test performance with pitching velocity using tests easy to implement, and derived an optimal set of parameters for predicting pitching velocity. We elected to not use kinematic or kinetic parameters because the required equipment is expensive and the corresponding measurements are usually conducted in laboratory settings [27]. Furthermore, kinematic and kinetic parameters represent pitching mechanics with little interpretative value for identifying a player's potential through physical tests. The outcome measures included in this study were shoulder $ER_{ROM}$ and $IR_{ROM}$, $CMJ_{loaded}$, $CMJ_{unloaded}$, $sprint_{30}$, age, weight, and height tests. These tests are easy to implement and were executed several times in different baseball stadiums. A group of 60 players could complete all tests within 2 hours, and the total cost of the equipment we used was less than USD $4,500. It is possible to run the accessments with equipment under USD $1,000 total,inclding **My Jump 2** APP [28,29], and Dashr timing system [30],. Our results were determined to be correlated with pitching velocity (adjusted $R^2 = 0.230$, $p < 0.001$). Among age, height, weight, $ER_{ROM}$, $IR_{ROM}$, $sprint_{30}$, $sprint_{ratio}$, $CMJ_{loaded}$, $CMJ_{unloaded}$, and $CMJ_{ratio}$, only height, $sprint_{ratio}$, and $CMJ_{ratio}$ were correlated with pitching velocity. Height is an inherent contributor to pitching velocity because taller pitchers have longer arms and can thus generate greater leverage power. Additionally, a greater height affords a larger room/path during the arm acceleration phase.

**Table 2. Coefficients with $p$ values for parameters.**

| Estimate | Std. | Error | t | p-value | Pr(>|t|) |
|---|---|---|---|---|---|
| (Intercept) | 11.489 | 32.137 | 0.358 | 0.722 | |
| height | 0.212 | 0.076 | 2.787 | 0.007 | ** |
| $CMJ_{ratio}$ | 67.141 | 33.513 | 2.003 | 0.050 | * |
| $CMJ_{unloaded}$ | 1.118 | 0.698 | 1.602 | 0.115 | |
| $CMJ_{loaded}$ | -1.290 | 0.807 | -1.599 | 0.115 | |
| $Sprint_{ratio}$ | 66.332 | 26.531 | 2.500 | 0.015 | * |

A positive $CMJ_{ratio}$ was derived in this study. This signifies that pitchers carrying a 20-kg load could jump to a similar height to that achieved when jumping without such a load. Kraska et. al. studied the relationship between strength characteristics and $CMJ_{ratio}$, and found the ability to produce higher peak and instantaneous forces and IRFD is related to jump height and $CMJ_{ratio}$. One of their findings is that stronger athletes have smaller decrements in vertical jump heights associated with weighted jumps compared with weaker athletes[16]. The study conducted by Aagaard et. al., showed that training produced increases in neural drive (IRFD) is associated with adaptations in the contractile strength of skeletal muscle [31]. Thus, pitchers with better $CMJ_{ratio}$ would demonstrate a relatively higher neural drive in the contractile skeletal muscle and lead to faster pitching velocity. Moreover, the $sprint_{ratio}$ coefficient was found to be positive; a higher $sprint_{ratio}$ value indicates a faster pitching velocity. A higher $sprint_{ratio}$ value actually indicates a longer first 10-meter, and shorter 10 to 30-meter sprint time. While 10-meter sprint is commonly tested for baseball players [13,32], there does not seem to be a consensus of which distance is the best one to test, thus 60-yard, 40-yard, and 20-yard have all been used by the MLB teams [33]. We chose 30-meter test so that in addition to the first 10 meters, 10 to 30-meter sprint time can also be measured to look at transitional acceleration. Coleman and Amonette found the time from Home-plate to First-Base is most affected by acceleltration pure acceleration [34]. Initially, sprinting time of 10-meter, 30-meter, and the $sprint_{ratio}$ are all included in our equation. But only $sprint_{ratio}$ is picked by the stepwise multiple linear regression analysis. This does not mean the first 10-meter sprint time is not important, but that for this group of athletes, transitional acceleration shows more significant correlation to pitching velocity. Maćkała et al. [32] reported that 10-m sprint time was significantly correlated with the stride index (SI, defined as sprint stride length/leg length); the SI has been suggested to be correlated with baseball pitching velocity [10,35]. However, $sprint_{30}$ was noted to be correlated with peak sprint velocity, maximum springing frequency, and SI [32]. In this study, $sprint_{ratio}$ was selected in the model because of its higher correlation with pitching velocity than 10-m and 30-m sprint time. This can be explained as transitional acceleration is more important to pitching velocity than pure acceleration, but we need more tests to verify this. Also we would like to test 10-meters and 30-meter saperately in the future, to make sure full effort spent from the athletes in the 10-meter test. Finally, the model revealed a moderate yet significant correlation between the field tests and pitching velocity for the group of amateur players recruited in this study. Accordingly, we suggest using the model only for players whose age and playing level are close to those included in this study.

A limitation of this study is that we analyzed only Asian adult male amateur players, and there are only limited number of tests we can try due to the time limit of the tryout. The generalizability of our findings to players of differet age, sex, playing level is unclear. We thus suggest that future studies further investigate possible tests those can predict players' potential and suitable for tryout events, and also try to apply the tests on players at other levels. As demonstrated in this study, quick field tests can be conducted to predict pitchers' potential. Pitchers who pitch at a velocity slower than the prediction can work on their pitching mechanics, and pitchers who outperform the model can focus on higher peak and instantaneous forces and isometric rates of force development to gain more velocity.

8 players were drafted after the tryout, and we track their measurements and field tests for the following year, along with their pitching velocity. We want to see how much room there is for players to grow with proper strength and conditioning training, both in range of motion and explosive power. We found some positive intra-individual correlations between the growth of these measurement results and the pitching velocity, and will keep following. Tryouts are very important for finding diamonds in the rough, but yet there are few researches on what are the best measurements and tests to be conducted in tryout events. We hope to work

on this topic not only the year of the tryout, but keep tracking the growth years after the players are drafted, to help more unpolished dimonds to be found, and shine.

## Supporting information

**S1 Dataset.**
(CSV)

## Author Contributions

**Conceptualization:** Jyh How Huang, Chih Hui Chiu.

**Data curation:** Szu-Hua Chen, Chih Hui Chiu.

**Formal analysis:** Chih Hui Chiu.

**Funding acquisition:** Jyh How Huang.

**Supervision:** Jyh How Huang.

**Writing – original draft:** Jyh How Huang, Szu-Hua Chen.

**Writing – review & editing:** Jyh How Huang, Chih Hui Chiu.

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
