## [Decision Letter · Decision Letter 0]

27 May 2021

PONE-D-21-08727

Correlation of pitching velocity to the anthropometric measurements for male adult baseball players

PLOS ONE

Dear Dr. Chiu,

Thank you for submitting your manuscript to PLOS ONE. After careful consideration, we feel that it has merit but does not fully meet PLOS ONE’s publication criteria as it currently stands. Therefore, we invite you to submit a revised version of the manuscript that addresses the points raised during the review process.

Please emphasize the novelty of the study, which both reviewers pointed. Please improve the clarify in methods/results. Thorough literature review as well as discussion are suggested as well.

We look forward to receiving your revised manuscript.

Kind regards,

Kei Masani

Academic Editor

PLOS ONE

Journal Requirements:

2. In your Data Availability statement, you have not specified where the minimal data set underlying the results described in your manuscript can be found. Regarding Data-sharing policy, it is unclear why 'No'.

PLOS defines a study's minimal data set as the underlying data used to reach the conclusions drawn in the manuscript and any additional data required to replicate the reported study findings in their entirety. All PLOS journals require that the minimal data set be made fully available. For more information about our data policy, please see http://journals.plos.org/plosone/s/data-availability.

'The funders had no role in study design, data collection and analysis, decision to publish, or preparation of the manuscript.'

6. We note you have included a table to which you do not refer in the text of your manuscript. Please ensure that you refer to Table 2 in your text; if accepted, production will need this reference to link the reader to the Table.

Reviewers' comments:

Reviewer's Responses to Questions

**Comments to the Author**

1. Is the manuscript technically sound, and do the data support the conclusions?

Reviewer #1: Partly

Reviewer #2: No

2. Has the statistical analysis been performed appropriately and rigorously? 

Reviewer #1: Yes

Reviewer #2: Yes

3. Have the authors made all data underlying the findings in their manuscript fully available?

Reviewer #1: No

Reviewer #2: No

4. Is the manuscript presented in an intelligible fashion and written in standard English?

Reviewer #1: No

Reviewer #2: No

5. Review Comments to the Author

Reviewer #1: The authors prevent a valuable contribution to the fields of scouting, performance and coaching. That being said, I do feel these results have been produced elsewhere (like at Driveline baseball). There could be much more thorough review of literature, and tempered recommendations at the end of the paper. I have some suggestions for the authors to further investigate, primarily, including their 9 removed athletes that didn't meet the professional standard. I would like to see the author's data as well.

Line Abstract - Pitching, not Pithing

Line 3-5: I understand the author’s intention - to find the relationship between mechanics and velocity, but this is so insufficient when it comes to understanding the kinematics associated with velocity. In fact, only one paper is referenced regarding mechanics, when it is widely studied how workload, fatigue, mechanics, and anthropometry/skill contribute to velocity. Recommend citing a review paper such as Seroyer, S. T., Nho, S. J., Bach, B. R., Bush-Joseph, C. A., Nicholson, G. P., & Romeo, A. A. (2010). The kinetic chain in overhand pitching: its potential role for performance enhancement and injury prevention. Sports health, 2(2), 135-146.

Line 11 - “the extent to pitching velocity has not been fully explored” - please re-construct sentence. This does not make sense.

Line 13 - identify what changes in mechanics would cause change in velocity

Line 16 - Furthermore, Driveline Baseball did a replication study on vertical jumps and the correlation to velocity. You do cite some other non-academic work, and I would recommend this being cited here.

https://www.drivelinebaseball.com/2016/09/examining-the-relationship-between-vertical-jump-force-and-velocity/

Line 22 - you are referencing Mercier here, could you please highlight the strength of the relationship in question? What was the r2 value?

Line 28 - what is your definition of “inexpensive field tests”?

Line 38 - would like to know which previous studies are cited for each test

Line 54 - Nine of the participants that were excluded from this study - did you have complete data for these athletes? Irrespective of their threshold for a professional standard, this could be a good data set to show the sensitivity of your model. If these athletes had very poor tests and the model predicted a very low velocity, this would only strengthen your model. Recommend these data be included.

Line 66 - was it the same experimenter recording ROM for each participant? If not, what was the inter-rater reliability?

Line 75 - see above comment. What is your definition of an inexpensive field test? How much was the Gym Aware hardware?

Line 93 - reference location of Rapsodo company

Line 99 - R step{stats} - is this an error, or is this how the Stats package is referenced?

Line 106 - may relationships in biological systems are non-linear. Did you explore any non-linear terms in your regression modelling? Why not?

Line 109 - many other studies have included BMI - would recommend this also be included in your model. While not perfect, it would give some context to large, muscular athletes.

Line 104 and Line 112 - Define VIF before using acronym

Line 132 - Total COST of the equipment at 4500, not COSE

Line 132 - 4500 USD is definitely less expensive than a full motion capture lab, but it isn’t inexpensive in my opinion - particularly if you’re looking at youth athletes.

Line 148 - what is the “first move when pitching”?

Line 160 - it sounds like you could erase the limitation of “not professional” by included the other 9 athletes who were not at the professional standard? Would like to see these data.

Line 162 - I think this section of the paper is very important for the layman, particularly in an open source journal. However, I think the examples you have given here are requiring more context. For example, your model has an RSS of 3.57. Your two examples have an error of 2 and 3 km/h. This is very close to the model error, and I don’t know if it’s a strong enough of an example, in the lack of other data on mechanics for you to make your claims. A generic example of a “mechanical tweak” is misleading - what kind of mechanical tweak are we considering? Furthermore, extensive research exists on the influence of fatigue on pitching velocity, as well as grip and the influence on spin rate, efficiency and velocity. These are all things that could lead to changes in velocity and are not controlled in this study.

Reviewer #2: The aim of study was to predict the outcomes that explain the pitching velocity in adult baseball pitchers. The measurements were age, height, weight, and some field tests including counter movement jump (CMJ), 20 kg-loaded CMJ, 10 m and 30 m sprint time, and maximum internal/external rotation angle of the throwing shoulder. The authors revealed that height, the ratio of the sprint time (10 m / 30 m), and the ratio of CMJ (unloaded / loaded) could predict pitching velocity. Pitching velocity is one of key factors to put out a butter for baseball pitchers and it is important to take the measure of the ability to throw a fastball for coaching staff, scout, and trainer and so on.

However, I have concerns on the followings; (1) a lack of novelty (previous studies have already reported similar results), (2) the use of low costs equipment and the ease of set up are insufficient as a claim of this study, (3) the rationale for the selection of measurements is insufficient, (4) the methodological descriptions of each measurement were not written in detail, and (5) the superficial discussion, especially about the ratio of loaded CMJ/ CMJ.

General comments

1. The authors’ selection on these measurements to predict pitching velocity are not justified: This manuscript does not mention the rationale that these measurements were appropriate to predict pitching velocity among field tests; and most of these measurements were already examined in the previous studies. Furthermore, the value of adjusted R2 was low, which does not fully support the authors’ conclusion. Therefore, unfortunately, it is difficult to find the novelty and relevance of this manuscript.

2. I suggest that this manuscript should examine whether each CMJ parameters (or speed test parameters) can explain pitching velocity. For example, as mentioned in the methods, CMJ and loaded CMJ were good indicator for the ability of explosive power exertion relate to pitching velocity. If each CMJ parameter relate to pitching velocity, CMJ and loaded CMJ tests were good assessment for pitching velocity. Thus, authors should examine the relationship between CMJ and pitching velocity in detail.

3. There are lack of clarities in descriptions of methods. For example, it is not clear whether the shoulder ROM was active or passive, and whether the posture during measurements was sitting or lying. Furthermore, the information about Rapsodo setup and number of pitches during trial or warming up are not described. Please revise the method section to provide much more details about the experiment.

Minor comments

1. The authors should space between values and unit.

2. The “throwing velocity” should be changed to “pitching velocity”.

3. References should be cited as “xxx et al. (20xx)…”.

4. Title: The title is not supported by results.

5. Intro: L15 lower extremity → upper extremity or throwing arm?

6. Methods: L49 The participants demographics such as their age, height, and weight were not provided.

7. Results: The results of two ratios about CMJ and sprint time were important factors that explain the pitching velocity. Thus, authors should describe the detailed analysis of CMJ and sprint time (e.g. height, time, and correlation between the values of ratio and pitching velocity).

8. Discussion: L132 cose → cost

9. References: 17 The journal is Res Q Exerc Sport?

6. PLOS authors have the option to publish the peer review history of their article (what does this mean?). If published, this will include your full peer review and any attached files.

Reviewer #1: **Yes: **Mike Sonne

Reviewer #2: No

---

## [Author Response · Author response to Decision Letter 0]

18 Aug 2021

PONE-D-21-08727

Correlation of pitching velocity to the anthropometric measurements for male adult baseball players

PLOS ONE

Dear Dr. Chiu,

Thank you for submitting your manuscript to PLOS ONE. After careful consideration, we feel that it has merit but does not fully meet PLOS ONE’s publication criteria as it currently stands. Therefore, we invite you to submit a revised version of the manuscript that addresses the points raised during the review process.

Please emphasize the novelty of the study, which both reviewers pointed. Please improve the clarify in methods/results. Thorough literature review as well as discussion are suggested as well.

We look forward to receiving your revised manuscript.

Kind regards,

Kei Masani

Academic Editor

PLOS ONE

Journal Requirements:

2. In your Data Availability statement, you have not specified where the minimal data set underlying the results described in your manuscript can be found. Regarding Data-sharing policy, it is unclear why 'No'.

PLOS defines a study's minimal data set as the underlying data used to reach the conclusions drawn in the manuscript and any additional data required to replicate the reported study findings in their entirety. All PLOS journals require that the minimal data set be made fully available. For more information about our data policy, please see http://journals.plos.org/plosone/s/data-availability.

The minimum dataset is uploaded and can be downloaded by all at

https://drive.google.com/file/d/1XOI_smCGZYf75pq1HzLrkgM9MlUI_7W5/view?usp=sharing

Reviewers' comments:

Reviewer's Responses to Questions

Comments to the Author

1. Is the manuscript technically sound, and do the data support the conclusions?

Reviewer #1: Partly

Reviewer #2: No

2. Has the statistical analysis been performed appropriately and rigorously?

Reviewer #1: Yes

Reviewer #2: Yes

3. Have the authors made all data underlying the findings in their manuscript fully available?

Reviewer #1: No

Reviewer #2: No

4. Is the manuscript presented in an intelligible fashion and written in standard English?

Reviewer #1: No

Reviewer #2: No

5. Review Comments to the Author

Reviewer #1: The authors prevent a valuable contribution to the fields of scouting, performance and coaching. That being said, I do feel these results have been produced elsewhere (like at Driveline baseball). There could be much more thorough review of literature, and tempered recommendations at the end of the paper. I have some suggestions for the authors to further investigate, primarily, including their 9 removed athletes that didn't meet the professional standard. I would like to see the author's data as well.

Line Abstract - Pitching, not Pithing

Corrected, thank you. We’ve also asked for help from professional English editing company and this manuscript is revised for proper style.

Line 3-5: I understand the author’s intention - to find the relationship between mechanics and velocity, but this is so insufficient when it comes to understanding the kinematics associated with velocity. In fact, only one paper is referenced regarding mechanics, when it is widely studied how workload, fatigue, mechanics, and anthropometry/skill contribute to velocity. Recommend citing a review paper such as Seroyer, S. T., Nho, S. J., Bach, B. R., Bush-Joseph, C. A., Nicholson, G. P., & Romeo, A. A. (2010). The kinetic chain in overhand pitching: its potential role for performance enhancement and injury prevention. Sports health, 2(2), 135-146.

Citation added, thank you.

Line 11 - “the extent to pitching velocity has not been fully explored” - please re-construct sentence. This does not make sense.

The sentence is re-constructed, thanks for reminding.

Line 13 - identify what changes in mechanics would cause change in velocity

We apologize for the overstatemen. The cited research is inter-individual, not intra-individual. Thank you for pointing it out.

The sentence is revised to “Studies have reported that better temporal parameters and kinematic parameters often infer to faster pitching velocity”

Line 16 - Furthermore, Driveline Baseball did a replication study on vertical jumps and the correlation to velocity. You do cite some other non-academic work, and I would recommend this being cited here.

https://www.drivelinebaseball.com/2016/09/examining-the-relationship-between-vertical-jump-force-and-velocity/

We do follow Driveline’s research and appreciate all the R&D done there. Citation added, thank you so much.

Line 22 - you are referencing Mercier here, could you please highlight the strength of the relationship in question? What was the r2 value?

Mercier gives no R2 value in the paper. That review is mainly about which research might be biased, and which parameters might truly be significant. 

We’ve added another reference, Relationship Between Performance Variables and Baseball Ability in Youth Baseball Players Nakata, Hiroki (2013), which has more detailed correlation R value for physical tests.

Line 28 - what is your definition of “inexpensive field tests”?

We were hoping that the total cost is under USD $500.

My jump2 APP for measure CMJ is USD $18, and shows good correlation.

Line 38 - would like to know which previous studies are cited for each test

Citations added after each test accordingly, thank you.

Line 54 - Nine of the participants that were excluded from this study - did you have complete data for these athletes? Irrespective of their threshold for a professional standard, this could be a good data set to show the sensitivity of your model. If these athletes had very poor tests and the model predicted a very low velocity, this would only strengthen your model. Recommend these data be included.

The raw data, including the 9 excluded participants, is uploaded and can be downloaded at

https://drive.google.com/file/d/1YO4NC2gGXitzGmlJaU3AhW2fMB6YSVvw/view?usp=sharing

We re-run the model with 9 excluded participants and answer it together in the Line 160 question.

Line 66 - was it the same experimenter recording ROM for each participant? If not, what was the inter-rater reliability?

To finish all the tests in 2 hours, there were 3 experimenter recording ROM. Though we haven’t tested the inter-rater reliability, but all of experimenter are trained by Taiwan Athletic Trainers’ Society, by the same instructor and passed the certification exam. 

Line 75 - see above comment. What is your definition of an inexpensive field test? How much was the Gym Aware hardware?

GymAware is USD $1,995. We were thinking that, comparing to Kistler force plate which costs over $40,000, under 2k is not expensive. It is still not cheap, we agree. 

We agree that “inexpensive” is probably not the best word to describe it, “affordable” is used instead, plus option for under $1000 for all is added in the paper.

Line 93 - reference location of Rapsodo company

Reference updated, thank you.

Line 99 - R step{stats} - is this an error, or is this how the Stats package is referenced?

In R programming, the name of the package is enclosed by {}. To better deliver the message, the text has been modified accordingly. “A linear stepwise multiple regression analysis was performed using R with step function in {stats} package to investigate……….”

Line 106 - may relationships in biological systems are non-linear. Did you explore any non-linear terms in your regression modelling? Why not?

The ways to fit a nonlinear model are infinite. Without knowing a strong foundation of the starting values for the nonlinear algorithm from the literature, we preferred taking a conservative approach and fitting a linear model, which gives an ability to better interpret the R square and p value of each dependent variable. 

Line 109 - many other studies have included BMI - would recommend this also be included in your model. While not perfect, it would give some context to large, muscular athletes.

Thank you for the suggestion. We ran an additional statistical analysis with BMI included in the regression model. The overall model fitting was not improved (R square = 0.223 versus 0.230 in the original model), thus we prefer keeping the model as it is (few predictors and higher R square value).

Fig1. The statistical result of the model presented in this study. (without BMI)

Fig2. The statistical result of the model with BMI included

Line 104 and Line 112 - Define VIF before using acronym

Thanks. We have edited the text accordingly. “Variance inflation factor (VIF) value was used to examine the multicollinearity”

Line 132 - Total COST of the equipment at 4500, not COSE

Corrected, thank you.

Line 132 - 4500 USD is definitely less expensive than a full motion capture lab, but it isn’t inexpensive in my opinion - particularly if you’re looking at youth athletes.

GymAware USD $1,995, can be replace by My Jump2 APP, which is USD $18 and has been validated by some research. https://147.197.131.104/handle/2299/21027.

A DIY laser gate measuring sprinting time costs around USD $120 can be used to measure sprinting time instead of Witty timing system. https://create.arduino.cc/projecthub/Pablerdo/wireless-laser-gate-timing-system-for-track-and-field-ba8cd9

or

Dashr timing system costs USD $350 https://www.dashrsystems.com/products

Line 148 - what is the “first move when pitching”?

Sentence is revised as follows, “In contrast to batting, the force does not have to be developed in the initial wind up move in pitching.”.

Line 160 - it sounds like you could erase the limitation of “not professional” by included the other 9 athletes who were not at the professional standard? Would like to see these data.

When drawing the tryout pitchers’ max velocity on a chart, there is a gap shown.

It looks like a good idea to make a cut at either 128km/h or 130km/h, since the data under 130km/h is very sparse. 

Anyways, we still re-run the model with 9 removed pitchers added to the dataset, and here is the result.

Adjusted R2 is 0.161. And with the 9 low velo pitchers removed

Adjusted R2 is 0.230, with all the coefficients being significant. 

Line 162 - I think this section of the paper is very important for the layman, particularly in an open source journal. However, I think the examples you have given here are requiring more context. For example, your model has an RSS of 3.57. Your two examples have an error of 2 and 3 km/h. This is very close to the model error, and I don’t know if it’s a strong enough of an example, in the lack of other data on mechanics for you to make your claims. A generic example of a “mechanical tweak” is misleading - what kind of mechanical tweak are we considering? Furthermore, extensive research exists on the influence of fatigue on pitching velocity, as well as grip and the influence on spin rate, efficiency and velocity. These are all things that could lead to changes in velocity and are not controlled in this study.

Yes, we agree that grip, fatigue, and lots of other factors affects the pitching velocity. We used too many assertions. After lots of discussions, we decided to remove the “practical applications” subsection and feel the quality of the paper will be better doing so.

Reviewer #2: The aim of study was to predict the outcomes that explain the pitching velocity in adult baseball pitchers. The measurements were age, height, weight, and some field tests including counter movement jump (CMJ), 20 kg-loaded CMJ, 10 m and 30 m sprint time, and maximum internal/external rotation angle of the throwing shoulder. The authors revealed that height, the ratio of the sprint time (10 m / 30 m), and the ratio of CMJ (unloaded / loaded) could predict pitching velocity. Pitching velocity is one of key factors to put out a butter for baseball pitchers and it is important to take the measure of the ability to throw a fastball for coaching staff, scout, and trainer and so on.

However, I have concerns on the followings; (1) a lack of novelty (previous studies have already reported similar results), (2) the use of low costs equipment and the ease of set up are insufficient as a claim of this study, (3) the rationale for the selection of measurements is insufficient, (4) the methodological descriptions of each measurement were not written in detail, and (5) the superficial discussion, especially about the ratio of loaded CMJ/ CMJ.

General comments

1. The authors’ selection on these measurements to predict pitching velocity are not justified: This manuscript does not mention the rationale that these measurements were appropriate to predict pitching velocity among field tests; and most of these measurements were already examined in the previous studies. Furthermore, the value of adjusted R2 was low, which does not fully support the authors’ conclusion. Therefore, unfortunately, it is difficult to find the novelty and relevance of this manuscript.

Yes, we agree most measurements were examined in previous studies, however, most previous studies are done in the lab setting with expensive equipment which not everyone has access to. We try to find the field tests possible to perform with mobile, affordable equipment in the short period of time so can be performed in most tryouts. The tests we performed can be done in a baseball field or any gym with 30 meters length and examine 60 athletes in 2 hours. R2 is mild, but p = 0.0003, so confidence level is very high, and the model is reliable.

In the study “Relationships between ball velocity and throwing mechanics in collegiate baseball pitchers” by Werner et. al, they conclude that “Body mass and 9 temporal and kinematic parameters related to pitching mechanics combine to account for 68% of the variance in ball velocity for a collegiate population of athletes.” That also says there are some factors affecting pitching velocity yet to be explored for pitching velocity. Thus, our model with inputs from field tests explains 23% of the variance in pitching velocity, seems reasonable and helps solve part of the puzzle. 

2. I suggest that this manuscript should examine whether each CMJ parameters (or speed test parameters) can explain pitching velocity. For example, as mentioned in the methods, CMJ and loaded CMJ were good indicator for the ability of explosive power exertion relate to pitching velocity. If each CMJ parameter relate to pitching velocity, CMJ and loaded CMJ tests were good assessment for pitching velocity. Thus, authors should examine the relationship between CMJ and pitching velocity in detail.

We did. It was explained in the Result section. Loaded CMJ and CMJ were still in the model after stepwise multi-regression. But they were eliminated in the multicollinearity testing process, meaning that they are highly correlated with the loaded CMJ/CMJ ratio and will boost the R2 falsely if included. 

3. There are lack of clarities in descriptions of methods. For example, it is not clear whether the shoulder ROM was active or passive, and whether the posture during measurements was sitting or lying. Furthermore, the information about Rapsodo setup and number of pitches during trial or warming up are not described. Please revise the method section to provide much more details about the experiment.

Shoulder ROM was passive, and posture was lying. The setup for Rapsodo and more warmup procedure are added to the Velocity sub-section, thank you for reminding. 

Minor comments

1. The authors should space between values and unit.

Revised, thanks for pointing it out.

2. The “throwing velocity” should be changed to “pitching velocity”.

Throwing velocity is changed to pitching velocity, thank you.

3. References should be cited as “xxx et al. (20xx)…”.

References updated, thank you.

4. Title: The title is not supported by results.

There is no one dominant factor in baseball pitching, A lot of factors add up to a pitcher’s velocity. We present that field tests correlate to a pitcher’s velocity at 23%, and the confidence level of the model is very high. 

5. Intro: L15 lower extremity → upper extremity or throwing arm?

It is lower extremity we refer to, and it is very important to pitching velocity. There were research on correlation of lower extremity and pitching velocity before and we cited two papers as well.

 [7] J. A. J. Guido and S. L. Werner, “Lower-Extremity Ground Reaction Forces in Collegiate Baseball Pitchers,” The Journal of Strength & Conditioning Research, vol. 26, no. 7, pp. 1782–1785, Jul. 2012, doi: 10.1519/JSC.0b013e31824e1211.

[8] “Relationship Between Vertical Jump Force and Pitching Velocity,” Driveline Baseball, Sep. 08, 2016. https://www.drivelinebaseball.com/2016/09/examining-the-relationship-between-vertical-jump-force-and-velocity/ (accessed Jun. 22, 2021).

6. Methods: L49 The participants demographics such as their age, height, and weight were not provided.

Since Taiwan is not very international regarding the athletes living here, participants in this tryout are very close in age, height, and weight. We’ve added that they are all Asian male in the sentence, thanks for reminding.

A total of 64 Asian male pitchers (age: 23.9 ± 2.8 years; height: 180.3 ± 5.9 cm, weight: 81.4 ± 10.9 kg)

7. Results: The results of two ratios about CMJ and sprint time were important factors that explain the pitching velocity. Thus, authors should describe the detailed analysis of CMJ and sprint time (e.g. height, time, and correlation between the values of ratio and pitching velocity).

We did explain it in the discussion section. On the other hand, we do not want to overstate it. We did present the result objectively with some hypothesis.

8. Discussion: L132 cose → cost

Corrected, thank you.

9. References: 17 The journal is Res Q Exerc Sport?

Corrected, thanks for pointing it out.

6. PLOS authors have the option to publish the peer review history of their article (what does this mean?). If published, this will include your full peer review and any attached files.

Do you want your identity to be public for this peer review? For information about this choice, including consent withdrawal, please see our Privacy Policy.

Reviewer #1: Yes: Mike Sonne

Reviewer #2: No

---

## [Decision Letter · Decision Letter 1]

8 Nov 2021

PONE-D-21-08727R1Correlation of pitching velocity to the anthropometric measurements for male adult baseball playersPLOS ONE

Dear Dr. Chiu,

Thank you for submitting your manuscript to PLOS ONE. After careful consideration, we feel that it has merit but does not fully meet PLOS ONE’s publication criteria as it currently stands. Therefore, we invite you to submit a revised version of the manuscript that addresses the points raised during the review process.

ACADEMIC EDITOR: Please follow the reviewer 2's comments 1 and 3, emphasizing the novelty and deepen the discussion. Also, please justify your method with considering the reviewer 2's comment 2.

We look forward to receiving your revised manuscript.

Kind regards,

Kei Masani

Academic Editor

PLOS ONE

Reviewers' comments:

Reviewer's Responses to Questions

**Comments to the Author**

1. If the authors have adequately addressed your comments raised in a previous round of review and you feel that this manuscript is now acceptable for publication, you may indicate that here to bypass the “Comments to the Author” section, enter your conflict of interest statement in the “Confidential to Editor” section, and submit your "Accept" recommendation.

Reviewer #2: All comments have been addressed

2. Is the manuscript technically sound, and do the data support the conclusions?

Reviewer #2: No

3. Has the statistical analysis been performed appropriately and rigorously? 

Reviewer #2: Yes

4. Have the authors made all data underlying the findings in their manuscript fully available?

Reviewer #2: No

5. Is the manuscript presented in an intelligible fashion and written in standard English?

Reviewer #2: Yes

6. Review Comments to the Author

Reviewer #2: General comments

1. The purpose of this study was to verify anthropometric measurements and field tests with inexpensive equipment to predict pitching velocity. However, this study is a lack of novelty because some previous studies (reference no. 9-12) already found the variables to predict pitching velocity, including height, jump performance, and sprint time. Indeed, although these previous studies examined the physical tests in the laboratory setting, but these measurements (excluding sprint test) can mostly execute in any places (e.g. laboratory, gymnasium, baseball field, and so on). Furthermore, some measurements in the previous studies (e.g. lateral to medial jump, medicine ball throw, and grip strength which could predict pitching velocity) were lower price than the measurements in this manuscript (e.g. Gymaware). Therefore, your claim that found the field tests with inexpensive equipment to predict pitching velocity is weak and a lack of novelty.

2. If the counter movement jump (CMJ) is a good indicator for predicting pitching velocity, authors should shift the direction of this study from to find variables that can predict pitching velocity using affordable device at field toward to verify how explosive power during CMJ contributes to the ball velocity in baseball pitchers. Although CMJ is different whole-body movement to baseball pitching, the ability of explosive force exertion during CMJ movement would relate to generate ball velocity in baseball pitchers. If so, authors can provide new methods to predict pitching velocity using only CMJ that previous studies did not yet demonstrate. Thus, authors should verify in detail the relationship the elements of explosive power during CMJ (and loaded CMJ) and pitching velocity (e.g. the rate of force development, EMG volume, kinematics, or kinetics).

3. Discussion of the current study is extremely cheap, and thus, authors should discuss more detailed, comparing to the finding of previous studies. For example, why did authors include passive shoulder internal/external rotation which was shown to not be a good predictor in the previous studies? Why did you choice the CMJ and not the lateral to medial jump which is considered a good predictor? Furthermore, why do differences appear in variables to predict pitching velocity among different age group (especially adolescent/youth baseball pitchers)? I would like you to properly discuss these things and the difference with the results of previous studies and meaning of selection of measurements in this study.

Minor comments

Introduction:

1. (L2) pitcher performance → pitching performance

2. (L11) thatbetter → that better

3. (L.56) Why did authors exclude the pitchers below 130 km/h? Authors should explain the reason why this exclusion criteria why you only included analysis above that. If authors included these subjects, the findings of this study might have shown different results.

4. (L88) Were subject's posture at the start of the sprint test uniform among all subjects? Because the time of first 10 m is strongly affected in acceleration at start, authors need to describe about this.

Discussion:

5. L141 The field test results → Our results

6. L163 velocitythan → velocity than

Table:

7. Is Table 2 going to be inserted somewhere within the manuscript?

7. PLOS authors have the option to publish the peer review history of their article (what does this mean?). If published, this will include your full peer review and any attached files.

Reviewer #2: **Yes: **Hirofumi Kobayashi

---

## [Author Response · Author response to Decision Letter 1]

21 Dec 2021

Reviewer #2: General comments

1. The purpose of this study was to verify anthropometric measurements and field tests with inexpensive equipment to predict pitching velocity. However, this study is a lack of novelty because some previous studies (reference no. 9-12) already found the variables to predict pitching velocity, including height, jump performance, and sprint time. Indeed, although these previous studies examined the physical tests in the laboratory setting, but these measurements (excluding sprint test) can mostly execute in any places (e.g. laboratory, gymnasium, baseball field, and so on). Furthermore, some measurements in the previous studies (e.g. lateral to medial jump, medicine ball throw, and grip strength which could predict pitching velocity) were lower price than the measurements in this manuscript (e.g. Gymaware). Therefore, your claim that found the field tests with inexpensive equipment to predict pitching velocity is weak and a lack of novelty.

Thank you for your comments. We made some major revisions and changed the paper title to "Correlation of pitching velocity with anthropometric measurements for adult male baseball pitchers in tryout settings". The purpose of this study is to find the suitable anthropometric measurements and field tests for time limited tryout events attended by several tens of players to find their potential of playing in the professional baseball league. Although most tryouts have some anthropometric measurements and field tests, but to our best knowledge, this study is the first one to report the correlation between the measurements, field tests and pitching velocity in the tryout settings. There are differences implementing measurements/tests in a lab setting and in tryouts, e.g., medicine ball throw. Most players are used to throw it to the wall at the angle parallel to the ground, but not shooting it for distance. It takes some practice throws for the athletes to find the optimal angle and posture but that takes time, which is luxury during tryouts while in the lab setting we might have time to instruct them, plus give them some time to practice. We did try medicine ball throw couple of times in other occasions, but found no significant correlation between medicine ball throw and pitching velocity without practice throws. We believe it will be a good test when most athletes are familiar with the move, but there really isn't enough time for the athletes to find the best posture and throwing angle during the tryout. We definitely agree lateral to medial jump is a good candidate test, but choose CMJ over it for the same reason, because of athletes' familiarities. Most athletes here are not familiar with lateral to medial jump, and for the first couple of attempts, some use their legs crossed/trunk twisted better than others to to gain some extra distance. So some players need practice jumps to be fair, which we didn't have time to do. We love the lateral to medial jump test and think it will be good test to implement in, say monthly tests on a pro team. But in a tryout, we picked CMJ for now for ease of implementation. Is lateral to medial jump is a better choice? We don't know until tested in tryouts. Grip strength was measured in the paper "Relationship Between Performance Variables and Baseball Ability in Youth Baseball Players" and found to be one of the significant predictors, but the age group is 6.4–15.7 years old, very different from ours. 

We appreciate your comments, which remind us to emphasize the value of our paper. We added "The measurements and tests in tryout settings should be easy to implement, take short time, do not need high skill levels, and correlate to the pitching velocity." in the abstract, along with other modifications and the changing of title.

2. If the counter movement jump (CMJ) is a good indicator for predicting pitching velocity, authors should shift the direction of this study from to find variables that can predict pitching velocity using affordable device at field toward to verify how explosive power during CMJ contributes to the ball velocity in baseball pitchers. Although CMJ is different whole-body movement to baseball pitching, the ability of explosive force exertion during CMJ movement would relate to generate ball velocity in baseball pitchers. If so, authors can provide new methods to predict pitching velocity using only CMJ that previous studies did not yet demonstrate. Thus, authors should verify in detail the relationship the elements of explosive power during CMJ (and loaded CMJ) and pitching velocity (e.g. the rate of force development, EMG volume, kinematics, or kinetics).

We didn't mean, and never said CMJ alone is a good indicator for predicting pitching velocity. CMJ is different movement to baseball pitching, just like grip strength test movement. But good grip strength and high amounts of lean body mass usually relate to jumping ability, which can hopefully be used to predict pitching velocity. The purpose of this study is to find the best combination tests to infer the potential of players, and we found that Height, CMJratio and SPRINTratio together can infer 23%. The stepwise multiple linear regression analysis actually picked CMJ weight to none weight ratio(CMJratio) over CMJ. The context is as following 

"In the stepwise multiple linear regression analysis, sprint10, weight, ERROM, IRROM, sprint30, and age were eliminated in this order. Of the remaining variables, CMJloaded, CMJunloaded, CMJratio, sprintratio, and height were selected on the basis of the sum of the squares. The variance inflation factor was used to test for multicollinearity, and CMJunloaded and CMJloaded were eliminated in the process. Therefore, the regression indicated an overall significant correlation of height, CMJratio, and Sprintratio with pitching velocity (F(4, 59) = 7.26, p < .001, adjusted R2 = .23), indicating that these parameters could explain approximately 23% of the variance in pitching velocity. Furthermore, examining individual predictors revealed that height (t = 2.82, p = .007), sprintratio (t = 2.98, p = .004), and CMJratio (t = 2.35, p = .022) were significant predictors in the model. The following model was thus derived for predicting pitching velocity. Velocity = 53.22 + 0.21 (height) + 14.3 (cmjWtoNW) + 75.03 (sprint10to30) + ϵ , adjusted R2 = 0.230, RSS = 3.57, p = 0.0003113"

The R square is for the whole equation, not CMJ alone. We executed the measurements/tests on 64 athletes, and presented the result honestly(p = 0.0003113). Our attempt is to find the measurements/tests suitable to be implemented in tryout events. This might not be the perfect combination, but we are hoping to get it started. 

3. Discussion of the current study is extremely cheap, and thus, authors should discuss more detailed, comparing to the finding of previous studies. For example, why did authors include passive shoulder internal/external rotation which was shown to not be a good predictor in the previous studies? Why did you choice the CMJ and not the lateral to medial jump which is considered a good predictor? Furthermore, why do differences appear in variables to predict pitching velocity among different age group (especially adolescent/youth baseball pitchers)? I would like you to properly discuss these things and the difference with the results of previous studies and meaning of selection of measurements in this study.

Passive shoulder internal/external rotation test in included because we are not just trying to correlate anthropometric measurements and field tests with current pitching velocity, we want to know, e.g., can pitchers with good shoulder ROM but weak strength gain more velocity than others if they are taken care by professional SnC trainers? 8 players were drafted after the tryout, and we track their measurements and field tests for the following year. We want to see how much room there is for a player to grow with proper strength and conditioning training, both in range of motion and explosive power. We have some findings but case number is not large enough, so we need some more years, more cases until we can present the findings. 

Thank you for suggesting adding "differences appear in variables to predict pitching velocity among different age group (especially adolescent/youth baseball pitchers)" to our discussion section. We checked couple other related research papers, including DEVELOPMENT OF A BASEBALL-SPECIFIC BATTERY OF TESTS AND A TESTING PROTOCOL FOR COLLEGE BASEBALL PLAYERS by KOHMURA et. al., and RELATIONSHIP BETWEEN PHYSICAL FITNESS AT THE END OF PRESEASON AND THE INSEASON GAME PERFORMANCE IN JAPANESE FEMALE PROFESSIONAL BASEBALL PLAYERS by WATANABE et. al. , and they focus on their participating group. None of them discuss the difference in variables for different age groups. 

We are aware that the development of strength in the muscles, hypertrophy, agility, etc are still growing for adolescent/youth baseball pitchers, and thank for your suggestion to how to add depth to our discussion. We've added more in depth discussion for possible reasons why CMJratio and SPRINTratio are picked, and it should be a lot better now. So please take another look at it, really appreciate.

Minor comments

Introduction:

1. (L2) pitcher performance → pitching performance

Changed, thank you.

2. (L11) thatbetter → that better

Edited, thanks.

3. (L.56) Why did authors exclude the pitchers below 130 km/h? Authors should explain the reason why this exclusion criteria why you only included analysis above that. If authors included these subjects, the findings of this study might have shown different results.

Those pitchers were not on the list. They were added last minute by some sponsors. Plus we do not see any pitchers who can survive with velocity below 130km/h in this league, except submarine pitchers.

Still, the raw data we uploaded for the paper have these 9 participants included should the readers are curious to try. 

https://drive.google.com/file/d/1YO4NC2gGXitzGmlJaU3AhW2fMB6YSVvw/view?usp=sharing

4. (L88) Were subject's posture at the start of the sprint test uniform among all subjects? Because the time of first 10 m is strongly affected in acceleration at start, authors need to describe about this.

All participants use standing start, added to that paragraph, thank you.

Discussion:

5. L141 The field test results → Our results

Modified, thank you.

6. L163 velocitythan → velocity than

That sentence is deleted during the revision process, thank you.

Thank you again for taking your time reviewing our paper, and Merry Christmas. :)

---

## [Editor Report · Decision Letter 2]

4 Mar 2022

Correlation of pitching velocity with anthropometric measurements for adult male baseball pitchers in tryout settings

PONE-D-21-08727R2

Dear Dr. Chiu,

We’re pleased to inform you that your manuscript has been judged scientifically suitable for publication and will be formally accepted for publication once it meets all outstanding technical requirements.

Kind regards,

Kei Masani

Academic Editor

PLOS ONE
---

## [Editor Report · Acceptance letter]

9 Mar 2022

PONE-D-21-08727R2 

Correlation of pitching velocity with anthropometric measurements for adult male baseball pitchers in tryout settings 

Dear Dr. Chiu:

I'm pleased to inform you that your manuscript has been deemed suitable for publication in PLOS ONE. Congratulations! Your manuscript is now with our production department. 

Kind regards, 

on behalf of

Dr. Kei Masani 

Academic Editor

PLOS ONE